# Therapeutic Potential of Bioactive Components from *Scutellaria baicalensis* Georgi in Inflammatory Bowel Disease and Colorectal Cancer: A Review

**DOI:** 10.3390/ijms24031954

**Published:** 2023-01-19

**Authors:** Jung Yoon Jang, Eunok Im, Nam Deuk Kim

**Affiliations:** Department of Pharmacy, College of Pharmacy, Research Institute for Drug Development, Pusan National University, Busan 46241, Republic of Korea

**Keywords:** inflammatory bowel disease, *Scutellaria baicalensis* Georgi, anti-inflammatory, colorectal cancer, molecular mechanisms

## Abstract

*Scutellaria baicalensis* Georgi (SBG), an herbal medicine with various biological activities, including anti-inflammatory, anticancer, antiviral, antibacterial, and antioxidant activities, is effective in treatment of colitis, hepatitis, pneumonia, respiratory infections, and allergic diseases. This herbal medicine consists of major active substances, such as baicalin, baicalein, wogonoside, and wogonin. Inflammatory bowel disease (IBD) comprises a group of inflammatory conditions of the colon and small intestine, with Crohn’s disease and ulcerative colitis being the main types. IBD can lead to serious complications, such as increased risk of colorectal cancer (CRC), one of the most common cancers worldwide. Currently, there is no cure for IBD, and its incidence has been increasing over the past few decades. This review comprehensively summarizes the efficacy of SBG in IBD and CRC and may serve as a reference for future research and development of drugs for IBD and cancer treatment.

## 1. Introduction

Inflammatory bowel disease (IBD), a type of chronic gastrointestinal inflammation, mainly includes Crohn’s disease (CD) and ulcerative colitis (UC) [1]. CD affects the small and large intestines, mouth, esophagus, stomach, and anus, whereas UC mainly affects the colon and rectum [2]. IBD is characterized by gastrointestinal discomfort, weight loss, bloody diarrhea, and periods of infiltration of neutrophils and macrophages, which release cytokines, proteolytic enzymes, and reactive species that cause ulceration and inflammation in the intestinal mucosa [3]. The prevalence and incidence of IBD increased remarkably during the second half of the 20th century, and, since the beginning of the 21st century, IBD is considered one of the most common gastrointestinal diseases as its incidence has increased in newly industrialized countries [4,5].

In the past, IBD was thought to occur primarily in white people in high-income countries, such as those of North America, Europe, and Australia. The incidence of IBD in these countries has leveled in recent years; however, its prevalence is still much higher than that in other countries. According to data from 2017, there were more than 6 million IBD cases worldwide, with nearly a quarter of them residing in the United States [6]. The highest prevalence of IBD was reported in Europe (UC 505 per 100,000 people in southeastern Norway; CD 322 per 100,000 people in Hessen, Germany) and North America (UC 286.3 per 100,000 people in Olmstead County, USA; CD 318.5 per 100,000 people in Nova Scotia, Canada) [4]. Patients with IBD have notably increased risk of colorectal cancer (CRC), primarily owing to the pro-neoplasmic effects of chronic intestinal inflammation [7,8,9]. According to a recent study, the risk of developing CRC in patients with IBD depends on the patient’s history of IBD and can increase by 2% after 10 years, 8% after 20 years, and 18% after 30 years [8,9]. However, owing to successful CRC surveillance programs and improved control of mucosal inflammation, its incidence has declined over the past 30 years. Thus, all major professional societies concur that IBD-associated CRC surveillance must occur regularly in the general population [9].

CRC, also known as colon or rectal cancer, is the third leading cause of cancer deaths in both men and women and has newly become the third most common cancer diagnosed in the United States [10]. According to South Korean projections for 2022, CRC ranks third in the expected mortality rate for men and second among women, and, in South Korea, it became the fourth most common cancer diagnosed in men and the third in women [11]. Currently, effective treatment methods for CRC include chemotherapy, surgery, radiation, or a combination of chemotherapy and radiotherapy. More than 50% of CRC patients who undergo surgical resection are cured, but 40–50% of these patients eventually relapse, and the likelihood of the patient returning to work is very low [12]. Drug resistance and chemotherapy-related toxic side effects are major causes of chemotherapy failure or discontinuation in CRC patients. As a result, there are problems in extending the survival period of CRC patients, reducing recurrence and metastasis rates, and improving quality of life, so alternative treatments are being sought [13].

Herbal plants have long been used to prevent and treat various diseases. Today, many people still rely on herbal nutraceuticals for primary healthcare. More than 50% of drugs currently in clinical use are of natural origin [14]. *Scutellaria baicalensis* Georgi (SBG) has been one of the most extensively used medicinal herbs in China for over 2000 years. The dried root of this plant has been used by the Chinese in traditional medicine called Scutellariae Radix or Huang-Qin and is officially listed in the Chinese Pharmacopoeia (2020) [15]. In addition, SBG is also listed in the European Pharmacopoeia (EP 9.0) and British Pharmacopoeia (BP 2018) [16]. Dried roots (Huang Qin) have a long history of medicinal use for treatment of inflammatory, gastrointestinal, and respiratory diseases [17]. A total of 56 flavonoids have been isolated from SBG, and the most representative components are baicalin, baicalein, wogonoside, and wogonin [18,19]. In this review, we present the effects of these components on IBD, IBD-associated CRC, and CRC.

## 2. IBD

### 2.1. CD and UC

IBD is caused by an interaction between genetic and environmental factors affecting the immune response [20]. Although the exact cause of IBD is unclear, the etiology is generally accepted as multifactorial and includes genetic predisposition, disorders of the gastrointestinal microbiota, dysfunction of the mucosal barrier, unregulated immune responses, and environmental and lifestyle factors [21,22].

CD can affect any part of the gastrointestinal tract, from the mouth to the anus, and usually affects individuals aged between 15 and 30 years. CD is discontinuous and can affect any layer of the intestinal wall. Patients with CD have symptoms that include abdominal fever, pain, bowel obstruction, or diarrhea with passage of mucus or blood [23].

UC is associated with persistent colonic inflammation, usually starting in the rectum and extending to the proximal portion of the colon, primarily affecting the inner epithelial layer of the colon [21]. It mainly affects adults aged 30–40 years and can weaken [24]. UC can be further classified into three types according to the degree of colonic involvement: proctitis, left-sided colitis, and diffuse colitis. Proctitis afflicts 30–60% of patients, and its symptoms include rectal bleeding, tension, and urgency. Left-side colitis affects 16–45% of patients with symptoms of proctitis, diarrhea, and abdominal cramps. The most severe form of UC is widespread colitis, affecting 15–35% of patients. This form can present with symptoms of left-sided colitis other than systemic symptoms, fatigue, and fever [23]. In addition to gastrointestinal symptoms, 25–40% of people with IBD may develop symptoms primarily associated with the skin, joints, eyes, and liver [25].

### 2.2. Onset of CRC in Patients with IBD

Although patients with IBD are at an increased risk of developing CRC, the overall incidence of IBD-associated CRC has declined in recent decades in Western countries. As demonstrated in previous studies, the risk of CRC in IBD increases with long-term, family history of CRC, coexisting primary sclerosing cholangitis, extent of colitis, and degree of inflammation [26]. However, the pathogenesis of IBD-associated CRC is not well understood. Similar to sporadic CRC, IBD-associated CRC results from sequential episodes of genomic changes [27]. Several interrelated pathways, including immune responses by mucosal inflammatory mediators, gut microbiota, and oxidative stress, are also associated with pathogenesis of IBD-associated CRC. Persistent colonic inflammation is a risk factor for CRC [26]. Sporadic CRC is defined by sequential histological and genetic changes, known as adenoma-carcinoma sequences. In contrast, IBD-associated CRC develops via the ‘inflammatory-dysplastic-carcinoma’ sequence [27,28,29]. Mutations contributing to pathogenesis of IBD-associated CRC are similar to those associated with sporadic CRC. However, multiple lines of evidence suggest that IBD-associated CRC may arise through different mechanisms of tumorigenesis compared with sporadic CRC. Loss of adenomatous polyposis coli (*APC*) occurs early in the development of sporadic CRC, whereas it is usually a late event in IBD-associated CRC. *TP53* mutations appear early in IBD-associated CRC, even before onset of dysplasia, but late in sporadic CRC [26,28,29,30].

### 2.3. IBD Treatment

The goal of IBD treatment is to reduce inflammation, which causes signs and symptoms. In the best case, IBD treatment can lead to symptom relief and reduced risk of long-term remission and complications [1]. Conventional treatment includes administration of aminosalicylates (sulfasalazine and 5-aminosalicylic acid), corticosteroids, immunomodulators (thiopurines, methotrexate, calcineurin inhibitors, and Janus kinase inhibitors), biologics such as pro-inflammatory cytokines (tumor necrosis factor-α (TNF-α), interleukin (IL)-12 and IL-23 inhibitors, and integrin antagonists), and, if necessary, other common measures or surgical resection [31].

However, these drugs have several drawbacks and adverse effects. Nonbiological treatments, such as aminosalicylates, thiopurines, and steroids, provide symptomatic improvement but do not alter the overall disease course in IBD [32]. Among immunomodulators, tacrolimus, a calcineurin inhibitor, can cause diabetes [33]. Biological therapies, including monoclonal antibodies targeting TNF-α, such as adalimumab, infliximab, golimumab, and certolizumab pegol, were introduced in the late 1990s to induce and maintain remission [34,35,36,37,38,39]. However, roughly one-third of patients with IBD are first-line non-responders to TNF-α inhibitor induction therapy [38,39,40,41], and the others become second-line non-responders during TNF-α inhibitor maintenance therapy [42,43,44]. Although the safety profiles of TNF-α inhibitors are generally acceptable, these drugs have side effects associated with infections and malignancies [45,46,47]. In addition, these biological therapies have the disadvantage of being expensive. As research intensifies, new therapies for treatment of IBD are emerging, focusing on apheresis therapy, stem cell transplantation, improvement of intestinal microecology, and exosome therapy [31]. These unapproved new therapies are often applied in research protocols but are limited by their unclear effects on IBD. Therefore, more studies, including long-term data, are needed to minimize the risk and optimize treatment outcomes. Identifying new drugs that are effective, economical, and less toxic has become a new goal.

### 2.4. Experimental Models of IBD

For decades, researchers have attempted to determine the causes and pathology of IBD. Researchers have developed many in vitro, ex vivo, and in vivo model systems to understand its causes and properly identify therapies [2].

#### 2.4.1. In Vitro and Ex Vivo IBD Models

In vitro cell line models included Caco-2, T84, and HT29 cell lines, organoids, Caco-2/HT29-MTX co-culture, and intestinal-immunological cultures [peripheral blood mononuclear cells (PBMCs), and macrophage cell lines from either human (THP-1) or murine origin (RAW 246.7)]. Caco-2 cells are an immortalized cell line of human colorectal adenocarcinoma [48]. Differentiated Caco-2 cells produce the same characteristics and specific digestive enzymes (e.g., peptidase and lactase) as small-bowel enterocytes, such as brush-border microvilli [49]. They can also produce various cytokines (IL-6, -8, and -15, TNF-α, and thymic stromal lymphopoietin) [50,51,52]. In addition, absorption-related studies are easy to perform [53,54,55]. HT29 cells are also human-colon-cancer-derived cells releasing cytokines similar to those released by Caco-2 cells [56]. HT29 cell lines produce similar cellular differentiation patterns and typical digestive enzymes (e.g., peptidase and lactase) to Caco-2 cells [57]. However, an important difference between Caco-2 and HT29 cell lines is that HT29 cells produce reasonably high levels of mucins [58].

Organoids are produced by pluripotent or adult stem cells. They are three-dimensional and can be differentiated into different cell types to perform related organ functions [59]. Organoids mimic the physiology of intestinal cells [60,61] and ecology of gut microbes [62,63]. They also enable investigation of the innate immune system [64]. Patient-derived organoids can also be used [65,66]. Researchers have found that exposure of fast-growing HT29 cells to high doses of methotrexate (MTX) results in development of mucus-secreting differentiated cells [67]. Therefore, the co-culture of Caco-2 and HT29-MTX can generate mucosa and comprise an excellent system to study microbial adhesion and colonic permeability [68]. This may also increase mucosal production [69]. Another cell line frequently used in intestinal research studies, the T84 cell line, contains microvilli on its surface and can develop into crypt-like cells when activated by modifying growth factors [70]. Similar to Caco-2 cells, T84 cells can rapidly develop into an absorptive layer of epithelial cells [71].

To simulate the interaction between intestinal and immune cells, researchers have used primary cells, such as PBMC, and THP-1 or RAW 246.7 macrophage cell lines, the main immune cells used to create various co-culture systems with intestinal cell lines. Human PBMCs are obtained from the peripheral blood of healthy donors or buffy coats and are used to study the effects of diet on immune cells [2]. The THP-1 cell line is a human leukemia cell line that was initially obtained from patients with acute monocytic leukemia. THP-1 cells are similar to macrophages and primary monocytes in terms of differentiation [72]. The RAW 246.7 cell line was obtained from mouse tumors using the Abelson leukemia virus. These cells can accurately simulate the intestinal environment and can express toll-like receptors that play an important role in pathogenesis of IBD [73].

Several ex vivo models have been used to overcome the aforementioned disadvantages of in vitro IBD models. The Ussing chamber evaluates ion transport in tissues, such as intestinal mucosa. It is used to study intestinal permeability and host–microbiome interactions using both animal and human tissues [74]. An inverted gut sac model was used to fully understand the kinetics and mechanisms of drug absorption; it has a mucous layer, intact intestinal tissue, and a large surface area [75]. Microfluidic gut-on-chip models have emerged as a new approach for studying gut function by integrating multiple cell types and the gut microbiome into one system [76]. This model enables co-culture of primary intestinal cells and macrophages; pluripotent human intestinal stem cells can also be used [77].

#### 2.4.2. In Vivo IBD Models

Experimental murine models of human IBD display immune pathological signatures similar to those of CD or UC [78]. These models include chemically induced (dextran sulfate sodium (DSS), trinitrobenzene sulfonic acid (TNBS), and oxazolone), spontaneous mutant [C3H/HeJBir (C3Bir) and SAMP1/Yit], and genetically engineered models (*IL-10^−/−^* knockout mice).

The DSS-induced colitis model is the most widely used colitis experimental murine model, in which DSS (molecular weight 40–50 kDa) is administered via drinking water [78]. DSS interrupts the epithelial barrier, exposes the lamina propria to luminal contents, and causes vascular and mucosal damage, subsequently resulting in activation of the inflammatory pathways [79]. The acute inflammatory response in DSS-induced colitis is represented by increased expression of IL-1β, IL-6, and TNF-α in the colon [78]. DSS-induced colitis is frequently used because of the simplicity of the administration process, ease of dose control to determine the severity of colitis, and adjustable duration to study the inflammatory or recovery process [80]. TNBS, used in chemically induced experimental models, is an oxidative nitroaryl compound administered intrarectally to animals to induce IBD. It causes colon damage that leads to necrosis associated with inflammation. The immune response is T helper type 1 (Th1)-mediated, characterized by infiltration of neutrophils, CD4^+^ T cells, and macrophages. Inflammation that spreads transversely results in transmural colitis [81]. Oxazolone, used in another chemically induced experimental model, is a hapten compound that causes acute colitis in animals when administered intrarectally in combination with ethanol. Oxazolone causes acute superficial mucosal inflammation in the distal colon, leading to a Th2-cell-mediated immune response with increased production of interleukins; the immune response type distinguishes this model from the TNBS-induced model, characterized by a Th1-mediated response [82].

Spontaneous mutant IBD mouse models develop IBD because of spontaneously occurring mutations but not gene suppression or transgenic overexpression. C3H/HeJBir (C3Bir) mice with a missense mutation in the third exon of the toll-like receptor 4 (TLR4) gene develop spontaneous inflammation in the cecum and colon. The SAMP1/Yit mouse variant showed elevated interferon-gamma and TNF-α levels, resulting in CD-like ileitis at approximately 10 weeks of age. Overall extension of stem cells in the crypt epithelium with severe colitis was observed in these mice [83].

Genetically engineered mouse models spontaneously develop colitis or ileitis. The best-known model is the IL-10^−/−^ knockout mouse, in which spontaneous colitis develops because of the inability of regulatory T (Treg) cells to produce IL-10. Specifically, inflammation of the colon is characterized by inflammatory infiltration of lymphocytes, macrophages, and neutrophils [82].

## 3. SBG

SBG, a flowering plant belonging to the Lamiaceae family, is widely distributed in China, Korea, Japan, Mongolia, and Russia [14]. Many scientific studies have demonstrated a wide range of pharmacological effects of SBG, including anti-inflammatory [84,85], antiviral [86], antibacterial [87], antioxidant [88], anticancer [89,90], and immunomodulatory activities [86]. The root of SBG, designated Scutellariae Radix or Huang Qin, is one of the most popular herbal medicines in China. SBG is particularly notable for its high flavonoid content. SBG contains four major flavones: baicalin (7-glucuronic acid, 5,6-dihydroxy-flavone), baicalein (5,6,7-trihydroxyfavone), wogonoside (wogonin-7-glucuronic acid), and wogonin (5,7-dihydroxy-8-methoxyflavone) at approximately 10.11%, 5.41%, 3.55%, and 1.3%, respectively, of the dry matter [91,92] (Figure 1A–D). Major aglycones baicalein and wogonin can be produced from glycosides baicalin and wogonoside, respectively [93]. Baicalin is hydrolyzed into baicalein by the action of the gut microbiota and reconverted back to baicalin through phase 2 metabolism in the liver [94,95]. Deglycosylation of wogonoside and its conversion to wogonin can be achieved via transformation of the enteric microbiome, enzyme catalysis, or herbal processing, such as soaking in water [96].

According to the Human Clinical Research Database, clinicaltrials.gov (accessed on 16 December 2022), a study on SBG was reported. NCT02828540 is a study of HT047, a complex herbal extract from the roots of *Pueraria lobata* and SBG. The main active compounds in HT047 are puerarin, daidzein, baicalin, baicalein, and wogonin. This study is the first human trial to explore the efficacy and safety of HT047 in patients with acute ischemic stroke. The results provided appropriate doses and evidence for the therapeutic effects of HT047 on stroke recovery [97]. Baicalin has been developed as a new drug (Huangqingan Tablets, manufactured by several companies, including Shanghai Hutchison Pharmaceuticals (Shanghai, China) and Jingfukang Pharmaceutical Group Co. Ltd., Chengde, China) to treat acute and chronic hepatitis. The total extract of the leaves and stems of SBG has also been developed as a new drug (Huangqin Jingye Jiedu Capsules) to treat sore throat [18].

## 4. Therapeutic Benefits of SBG

### 4.1. Effect of SBG on IBD

The anti-inflammatory effects of baicalin, baicalein, wogonoside, and wogonin have been reported. The effects of baicalin (Table 1), baicalein (Table 2), wogonoside (Table 3), and wogonin (Table 4) on IBD are summarized.

#### 4.1.1. Effect of SBG on COX-2

Cyclooxygenases-1 (COX-1) and COX-2 are antioxidant enzymes catalyzing conversion of free arachidonic acid to prostaglandin (PG) H2, which plays a significant role in inflammatory and pain responses. COX-1, first purified in 1976, is expressed in most tissues, whereas COX-2 was recently identified as a major inducible isoform producing high amounts of PGH2 [144]. COX-2 plays a major role in regulating intestinal inflammation. COX-2 is primarily expressed at sites of inflammation. Patients with active IBD have increased COX-2 expression levels in colonic epithelial cells, the myenteric plexus, and the medial layer of arteries [145]. In addition, COX-2 levels in blood samples of patients with active UC were significantly higher than those in healthy people and patients with palliative UC [1].

According to a study by Cui et al. [104], COX-2 expression was markedly increased by lipopolysaccharide (LPS) treatment in RAW 264.7 cells and decreased by baicalin treatment. In addition, COX-2 levels decreased in TNBS-induced UC rats compared to those in the TNBS-treated group, and it can also down-regulate pro-inflammatory mediators in the colonic mucosa. A decrease in pro-inflammatory cytokine production correlated with a decrease in mucosal TLR4 protein expression. The expression of p-nuclear factor kappa B (NF-κB) p65 protein was notably decreased, which was related to a similar decrease in p-IκB protein.

Several IBD model studies have shown that baicalein exhibits anti-inflammatory activity via COX-2 inhibition [85,118,128]. According to a study by Zhong et al. [85], baicalein significantly reduced the severity of DSS-induced colitis. Baicalein inhibited expression of COX-2 and inducible nitric oxide synthase (iNOS) and attenuated activity and phosphorylation of IKKβ and subsequent degradation of IκBα. In addition, baicalein inhibited p65 phosphorylation and nuclear translocation, decreased the DNA-binding activity of NF-κB, and inhibited the signal transducer and activator of transcription 3 (STAT3) phosphorylation and cyclin D1 expression.

According to a study by Yang et al. [133], wogonoside treatment of LPS-induced RAW 264.7 cells not only reduced production of inflammatory mediators, including nitric oxide (NO) and prostaglandin E2 (PGE2), but also prevented release of pro-inflammatory cytokines, including TNF-α and IL-6. In addition, wogonoside significantly inhibited gene expression of COX-2, iNOS, TNF-α, and IL-6.

Similarly, wogonin has been found to have anti-inflammatory effects in IBD models by reducing COX-2 [134,140,141,143]. Chi et al. [140] demonstrated that wogonin inhibited NO and PGE2 production in LPS-induced RAW 264.7 cells. They also showed that wogonin inhibited iNOS and COX-2 induction. Thus, they demonstrated that wogonin is not only a direct COX-2 inhibitor but also an inhibitor of iNOS and COX-2 induction.

#### 4.1.2. Effect of SBG on TLR4/NF-κB

TLR4 is a TLR receptor activated by bacterial LPS. Once activated, TLR4 up-regulates expression of genes that stimulate downstream mitogen-activated protein kinases and NF-κB signaling pathways and encode inflammatory cytokines [146]. Transcription factor NF-κB plays an important role in regulation of many genes encoding inflammatory mediators. NF-κB activation entails rapid phosphorylation of IκB in response to pro-inflammatory stimuli [147]. Free NF-κB produced during this process translocates to the nucleus and induces transcription of COX-2, iNOS, IL-1β, TNF-α, and PGE2 mRNAs. Thus, inhibition of NF-κB signaling has potential therapeutic benefits in treatment of inflammatory diseases [128,148]. Under normal physiological conditions, TLR4 expression is low in intestinal epithelial cells, resulting in mucosal integrity and protection against invading bacteria. However, its expression is up-regulated in IBD, leading to inflammation and tissue damage [149]. Additionally, patients with active UC have high TLR4 expression in intestinal epithelial cells [150]. Activation of the TLR4 signaling pathway induces intestinal myeloid cell infiltration and release of inflammatory mediators in the colonic tissue. Therefore, inhibition of this pathway is an effective strategy for treating IBD [151].

Numerous studies have demonstrated the anti-inflammatory effects of baicalin through TLR4/NF-κB inhibition in IBD models [84,98,99,103,104,107,108,114]. According to the study by Yan et al. [114], treatment of LPS-induced RAW 264.7 cells with baicalin resulted in effective suppression of the expression of inflammatory factors (TNF-α, IL-1β, IL-6, COX, and iNOS). Previous studies have reported that up-regulation of miR-181b is associated with endotoxin resistance of RAW 264.7 cells and restraint of the excessive immune response induced by LPS [152]. In addition, HMGB1 (high-mobility group box 1) affects the inflammatory response by binding to the TLR4 protein [153]. Thus, baicalin could prevent LPS-induced inflammation in RAW 264.7 cells through the miR-181b/HMGB1/TRL4/NF-κB pathway.

Several studies have shown that baicalein reduces TLR4/NF-κB, leading to anti-inflammatory effects [85,107,118]. According to a study by Luo et al. [118], expression of several pro-inflammatory mediator genes, including iNOS, intercellular adhesion molecule-1 (ICAM-1), monocyte chemotactic protein-1 (MCP-1), COX-2, TNF-α, and IL-1β, was significantly reduced in the colon tissues of TNBS-colitis mice. Baicalein attenuated TNBS-induced colitis, at least in part, by inhibiting the TLR4/MyD88 signaling cascade (NF-κB and mitogen-activated protein kinase (MAPK)) and inactivating nucleotide-binding oligomerization domain-like receptor pyrin domain containing 3 (NLRP3) inflammasome.

Wogonoside has anti-inflammatory effects via TLR4/NF-κB reduction [131,132]. Sun et al. [131] demonstrated that wogonoside can exert an anti-inflammatory effect through dual inhibition of NF-κB and NLRP3 inflammasome in DSS-induced murine colitis. In addition, wogonoside significantly reduced production of IL-1β, TNF-α, and IL-6 and inhibited activation of NF-KB and NLRP3, thus inhibiting the mRNA expression of proIL-1β and NLRP3 inflammasom in porbol-myristate-acetate-differentiated monocomposite THP-1 cells.

Several studies have reported that wogonin has anti-inflammatory effects through TLR4/NF-κB reduction [134,136,143]. According to a study by Zhou et al. [134], wogonin reduced oxidative damage and elevation of inflammatory mediators in DSS-induced UC mice. Wogonin also inhibits NF-κB by down-regulating COX-2 and iNOS, modulating the Nrf2 signaling pathway, and reducing TLR-4/NF-κB activation. In addition, wogonin induced apoptosis by suppressing B-cell lymphoma-2 (Bcl-2) and increasing Bcl-2-associated X protein (Bax) and caspase-3 and -9 expression.

### 4.2. Effect of SBG on CRC

Several studies have demonstrated that SBG inhibits CRC growth both in vitro and in vivo. We summarized the results of several studies on the effects of SBG, baicalin (Table 1), baicalein (Table 2), wogonoside (Table 3), and wogonin (Table 4) on CRC.

#### 4.2.1. Effect of SBG on Apoptosis in CRC

One way to treat cancer is to control or terminate the uncontrolled growth of cancer cells. Use of a cell’s own death mechanism is a very effective method. Apoptosis is a natural mechanism of programmed cell death [154]. It is a highly regulated process that eliminates unnecessary or unwanted cells [155]. There are two different apoptotic pathways: intrinsic and extrinsic, involving different types of signals [156]. The intrinsic apoptotic pathway uses mitochondria and mitochondrial proteins. The intrinsic pathway is regulated by the Bcl-2 family of proteins [157]. Many apoptotic stimuli result in up-regulation of BH3-only proteins, thereby activating both Bax and Bak. Bax is controlled by tumor suppressor gene *p53* [158]. Extrinsic or death-receptor-mediated pathways are initiated by binding of death-inducing ligands to cell surface death receptors. Membrane death receptors belong to the TNF receptor superfamily and include tumor necrosis factor-receptor 1 (TNF-R1/DR1), death receptor-4 (DR4) and -5 (DR5), and Fas (Apo-1/CD95/DR2). These receptors are activated by specific ligands, such as Fas ligand (FasL), TNF-α, and TNF-related apoptosis-inducing ligand (TRAIL) [157].

Baicalin inhibits growth of several types of human CRC cells and induces apoptosis [109,110,111,112]. According to a study by Yang et al. [110], baicalin significantly inhibited the viability of CRC (RKO and HCT116) cells. In CRC cells, baicalin treatment induced cell cycle arrest in the G1 phase, promoted p53-independent apoptosis, and prevented both exogenous and endogenous TGFβ1-induced epithelial–mesenchymal transition in CRC cells by inhibiting the TGF-β/Smad pathway. In vivo experiments confirmed the antitumor effects by down-regulating marker protein levels of the cell cycle, epithelial–mesenchymal transition (EMT), and stemness in orthotopically transplanted tumors of CRC cells in BALB/c nude mice.

Baicalein is cytotoxic and induces apoptosis in human CRC cells [89,119,120,123,124]. Sui et al. [123] have shown that baicalein inhibited proliferation of CRC (HCT116, A549, and Panc-1) cells in a dose-dependent manner. Decidual protein induced by progesterone (DEPP) was originally known as a progesterone-induced gene in a human endometrial stromal cell cDNA library [159]. Baicalein induced up-regulation of DEPP and GADD45A, resulting in a marked apoptotic response in human CRC cells by promoting activation of MAPKs, with a positive feedback loop between GADD45A and c-Jun N-terminal kinase (JNK)/p38.

Wogonoside inhibits human CRC cell growth and induces apoptosis [90,132]. According to a study by Han et al. [90], wogonoside prevented cell growth and induced mitochondria-mediated autophagy-related apoptosis in human LOVO CRC cells by controlling the phosphoinositide 3-kinase (PI3K)/AKT/mammalian target of rapamycin (mTOR)/p70S6K signaling pathway.

Wogonin induces apoptosis in in vitro and in vivo CRC models [96,124,135,137]. Feng et al. [135] have shown that wogonin exerts anti-colon cancer activity in AOM/DSS animal and HCT116 cell models. Wogonin induced apoptosis in HCT116 cells via increased endoplasmic reticulum (ER) stress. Excessive ER stress promoted cytoplasmic localization of p53 by increasing the phosphorylation of p53 at the S315 and S376 sites, induced caspase-dependent apoptosis, and inhibited autophagy.

#### 4.2.2. Effect of SBG on PI3K/AKT/mTOR in CRC

PI3Ks are intracellular lipid kinases implicated in cell proliferation, differentiation, and survival. The PI3K/AKT/mTOR signaling cascade promotes CRC oncogenesis in multiple ways, such as induction of drug resistance, inhibition of apoptosis, angiogenesis, metastasis, and EMT [160]. Overexpression of PI3K/AKT/mTOR signaling has been reported in diverse forms of cancer, particularly CRC [161]. Therefore, targeting PI3K/AKT/mTOR may be an important approach for cancer prevention and treatment [162].

Baicalin inhibits growth of CRC cells through PI3K/AKT/mTOR signaling [102,110,113]. According to a study by Zhu et al. [113], baicalin not only significantly inhibited cell viability, proliferation, migration, invasion, and angiogenesis but also promoted apoptosis in SW620 CRC cells. However, following inhibition of the PI3K/AKT/GSK-3β pathway using PI3K inhibitor LY294002, baicalin showed no significant inhibitory effects on the biological behavior of CRC, except for angiogenesis. Baicalin significantly inhibited phosphorylation of PI3K, AKT, and GSK-3β proteins and reduced mRNA levels. LY294002 further inhibited protein phosphorylation but did not significantly affect the mRNA. These effects of baicalin may be related to its role in prevention of the PI3K/AKT/GSK-3β pathway.

Baicalein also exhibits anticancer activity via the AKT pathway in in vitro and in vivo models of CRC [124,125]. Kim et al. [124] revealed that baicalein caused apoptosis through increased cell cycle arrest in the G1 phase in HT29 CRC cells. In addition, baicalein inactivated PI3K/AKT in a dose-dependent manner. Furthermore, administration of baicalein to mice inhibited growth of HT29 xenografts. Thus, baicalein can induce apoptosis through AKT inactivation in a p53-dependent manner in HT29 CRC cells.

Several studies have suggested that wogonoside exerts anticancer activity through the PI3K/AKT/mTOR pathway in CRC models [90,132]. According to a study by Sun et al. [132], wogonoside notably decreased disease severity, lowered tumor incidence, and suppressed development of colorectal adenomas in an AOM/DSS mouse model. In addition, wogonoside prevented inflammatory cancer cell proliferation and cell infiltration at the tumor site. In conclusion, wogonoside attenuates colitis-related tumor development in mice by inhibiting NF-κB activation through the PI3K/AKT pathway and inhibits progression of human CRC in an inflammatory microenvironment. It is a promising therapeutic agent for CRC.

Wogonin exerts anticancer activity via the PI3K/AKT pathway in CRC models [124,136,139]. For example, Tan et al. [139] have shown that wogonin inhibited growth of SW48 CRC cells via autophagy and apoptotic cell death. Wogonin induced SW48 cell arrest at the G2/M checkpoint of the cell cycle and inhibited the PI3K/AKT and STAT3 signaling pathways.

## 5. Conclusions

In this review, we outlined the current understanding of the mechanisms of the anti-inflammatory effects of the major constituents of SBG—baicalin, baicalein, wogonoside, and wogonin—in IBD and their anticancer efficacy in CRC. Traditional Chinese medicine has attracted significant attention in recent years because of its effectiveness in treatment of human diseases. Natural drugs have many advantages over conventional drugs, including better accessibility, economic feasibility, and low toxicity. The main constituents of SBG reduced release of inflammatory factors in several in vitro and in vivo models of IBD and showed anticancer activity by regulating different pathways. Therefore, SBG may represent a promising drug for IBD, and its components may be important lead compounds for development of new chemopreventive agents against IBD-associated CRC and CRC. However, although the potential of these compounds has been proven in various in vitro and in vivo settings, more preclinical and clinical studies are needed to validate their potency and anticancer activity.

## Figures and Tables

**Figure 1 ijms-24-01954-f001:**
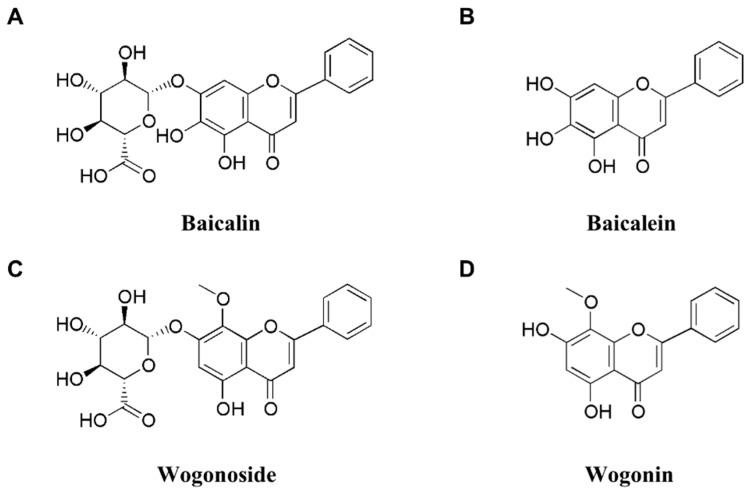
Chemical structures of baicalin (**A**), baicalein (**B**), wogonoside (**C**), and wogonin (**D**).

**Table 1 ijms-24-01954-t001:** The effect of baicalin on inflammatory bowel disease and colorectal cancer.

Disease	Animal/Cell Models	Dose and Times	Up-Regulation	Down-Regulation	Refs.
*UC*	DSS-induced UC mice	100 mg/kg, 5 days		CD14, IL-6	[84]
		150 mg/kg/d, 30 days	IκB-α	MPO, NO, IL-1β, IL-6, TNF-α, IL-33, NF-κB p65, p-NF-κB p65, p-IκB-α	[98]
		100 mg/kg, twice daily for 7 days	IL-10	TLR2, TLR4, TLR9, NF-κB p65, IL-6, IL-13, TNF-α,	[99]
		20 mg/kg, 13 days	CDX2, MDR1 (colon), Cyp3a11 (colon)	TNF-α, IL-6	[100]
	TNBS-induced UC rat	100 mg/kg, every 2 days for 14 days	ZO-1, occludin, SOD, GSH, IL-10, Foxp3, acetic acid (fecal), propionic acid (fecal), butyric acid (fecal)	MUC2, ROS, MDA, Th17/Treg, IL-17, RORγt	[101]
		100 mg/kg/d, 14 days	IL-10, ZO-1, β-catenin	p-PI3K/PI3K, p-AKT/AKT, TNF-α, IL-6, IL-1β	[102]
		90 mg/kg, 4 weeks	SOD, CAT, GSH-Px	MDA, IL-1β, MPO, PGE2, TNF-α, caspase-3 and -9, Bcl-2/Bax, Cyt *c*, NF-κB p65, p-IKKβ/IKKβ, p-IKBα/IKBα	[103]
		4 mL baicalin liquid (5 mg/mL), 15 days		MPO activity, ICAM-1, MCP-1, COX-2, TNF-α, IL-1β, IL-6	[104]
		120 mg/kg, 15 days	CAT, GSH-Px, SOD	MDA, apoptosis rate, ROS	[105]
		10 mL/kg, 7 days	Foxp3	MPO, TNF-α, IL-1β, IL-12, IFN-γ, IL-17, IL-6, IL-10, TGF-β, RORγt	[106]
	TNBS-induced UC rat (intestinal mucosal cell)	100 mg/kg/d, 14 days	Bcl-2	caspase-3 and -9, FasL, Bax	[102]
	UC rats of various complex factors	100 mg/kg, 7 days	SOD, ATP	AST, ALT, GGT, TG, MDA, NO, cAMP/cGMP, IL-6, IL-1β, IL-17, NF-κB p65, p38	[107]
*CRC*	HT29 cells (LPS-treated)	50 μM, 2 or 24 h		CD14, MyD88, NF-κB p65	[84]
		100 ng/mL, 30 min	IL-10, Bcl-2, ZO-1, β-catenin	p-PI3K/PI3K, p-AKT/AKT, TNF-α, IL-6, IL-1β, caspase-3 and -9, FasL, Bax	[102]
		10 μg/mL, 30 min	LC3, ATG5, BECN1, claudin 1	TNF-α, IL-1β, NF-κB p65	[108]
	HT29 cells	150 μM, 48 h	cleaved of caspase-3, apoptosis, HIC1, PDCD4, PTEN, E2F2, E-cadherin	miR-10a, miR-23a, miR-30c, miR-31, miR-151a, miR-205, Bcl-2, c-Myc	[109]
	RKO and HCT116 cells	100 μg/mL, 48 h	cleaved of caspase-3, -8, -9, and PARP-1, E-cadherin, Cytokeratin 18, Claudin 1, Smad7	XIAP, NF-κB, survivin, Bcl-2, p-AKT(Ser473), cyclin D1, cyclin E1, cyclin B1, N-Cadherin, vimentin, snail, slug, twist, TGF-β1, p-Smad2/3, Smad2/3/4, CD133, CD44, SOX2, OCT4, Nanog	[110]
	SW480 cells	400 μg/mL, 24 or 48 h	apoptosis, cleaved of caspase-3 and PARP-1	Sp1	[111]
	SW620 cells	200 µM, 24 h	sub-G1, caspase-3, -8, and -9 activity, ROS		[112]
		50 μmol/L, 48 h		p-PI3K, p-AKT, p-GSK-3β	[113]
	HCT116, SW480, and HT29 cells	20 µM, 72 h	apoptotic cells, p-ERK, p-p38		[89]
	HCT116 xenograft	200 mg/kg, 32 days	E-cadherin, cleaved of caspase-3 and PARP-1	Ki67, vimentin, N-cadherin, CD44, CD133, cyclin D1, cyclin B1	[110]
	HT29 xenograft	100 mg/kg, 3 weeks	cleaved of caspase-3, miR-204	Ki67, c-Myc, miR-10a, miR-23a, miR-30c, miR-31, miR-151a	[109]
*Miscellaneous*	RAW 264.7cells (LPS-treated)	50 μM, 2 or 24 h		NO, IL-6, MHC II, MyD88, NF-κB p65, CD14	[84]
		1.0 μmol/L, 24 h		HMGB1, TLR4, p-IκBα, p-p65, p65 (nuclear), IL-1β, IL-6, COX, iNOS	[114]
				p-p65/p-65, p-IKBα/IKBα	[103]
		1.0 × 10^−5^ M	CAT, GSH-Px, SOD, Bcl-2	MDA, apoptosis rate, TGF-β1, caspase-3 and -9, cleaved caspase-3 and -9, Bax, Fas, FasL	[105]
		5 × 10^−4^ M	IκB	TLR4, NF-κB, p-NF-κB, p-IκB, ICAM-1, MCP-1, COX-2, TNF-α, IL-1β, IL-6	[104]
	LPS-treated mice	100 mg/kg (i.p.), 3 days		CD14, IL-6	[84]
	Mouse peritoneal macrophages (LPS-treated)	50 μM, 24 h	IL-10, Arg-1, IRF4	TNF-α, IL-23, IRF5, M1/M2 macrophage ratio	[115]

AKT, protein kinase B; ALT, alanine aminotransferase; Arg-1, arginase-1; AST, aspartate aminotransferase; ATG5, autophagy-related gene 5; ATP, adenosine triphosphate; Bax, Bcl-2-associated X protein; Bcl-2, B-cell lymphoma 2; cAMP, cyclic adenosine monophosphate; CD14, cluster of differentiation 14; CDX2, caudal type homeobox 2; cGMP, cyclic guanosine monophosphate; CAT, catalase; COX, cyclooxygenase; CRC, colorectal cancer; Cyt *c*, cytochrome *c*; DSS, dextran sulfate sodium; ERK, extracellular signal-regulated kinase; FasL, Fas Ligand; Foxp3, forkhead box p3; GGT, γ-glutamyl transpeptidase; GSH, glutathione; GSH-Px, glutathione peroxidase; HIC1, hypermethylated in cancer 1; HMGB1, high-mobility group box 1; IFN-γ, interferon gamma; IκBα, nuclear factor of kappa light polypeptide gene enhancer in B-cells inhibitor alpha; ICAM-1, intercellular adhesion molecule-1; IKKβ, inhibitor of nuclear factor kappa-B kinase subunit beta; IL-, interleukin; iNOS, inducible nitric oxide synthase; i.p., intraperitoneal injection; LC3, microtubule-associated protein 1A/1B-light chain 3; LPS, lipopolysaccharide; MCP-1, monocyte chemoattractant protein 1; MDA, malondialdehyde; MDR1, multidrug resistance protein 1; MyD88, myeloid differentiation factor 88; MPO, myeloperoxidase; NF-κB p65, nuclear factor kappa-light-chain-enhancer of activated B cells p65; NO, nitric oxide; MHC II, major histocompatibility complex class II; HMGB1, high mobility group box 1; p-, phosphorylation; OCT4, octamer-binding transcription factor 4; PARP, poly(ADP-ribose) polymerase; PDCD4, programmed cell death protein 4; PGE2, prostaglandin E2; PI3K, phosphoinositide 3-kinase; PTEN, phosphatase and tensin homologue deleted on chromosome ten; ROS, reactive oxygen species; RORγt, retinoic acid-related orphan receptor gamma t; SOD, superoxide dismutase; SOX2, sex determining region Y-box 2; TG, triglyceride; TGF-β1, transforming growth factor beta-1; TNF-α, tumor necrosis factor α; TLR, toll-like receptor; TNBS, trinitrobenzene sulphonic acid; UC, ulcerative colitis; XIAP, X-linked inhibitor of apoptosis protein; ZO-1, zonula occludens-1.

**Table 2 ijms-24-01954-t002:** The effect of baicalein on inflammatory bowel disease and colorectal cancer.

Disease	Animal/Cell Models	Dose and Times	Up-Regulation	Down-Regulation	Refs.
*UC*	DSS-induced UC mice	25 mg/kg, 7 days	IκBα	COX-2, iNOS, cyclin D1, p-IκBα, p-p65, p65, IKKβ, NF-κB, p-STAT3, STAT3	[85]
		20 mg/kg, 10 days	IL-4	IFN-γ	[116]
		40 mg/kg, 10 days	CYP1A1	IL-6, IL-10, IL-17, IL-22, TNF-α, TNF-β	[117]
		20 mg/kg, 13 days	CDX2, MDR1 (colon), Cyp3a11 (colon)	TNF-α, IL-6	[100]
	UC rats of various complex factors	100 mg/kg, 7 days	SOD, ATP	AST, ALT, GGT, TG, MDA, NO, cAMP/cGMP, IL-6, IL-1β, IL-17, NF-κB p65, p38	[107]
*CD*	TNBS-induced CD mice	20 mg/kg/d, 10 days		iNOS, ICAM-1, MCP-1, COX-2, TNF-α, IL-1β, TLR4, MyD88, p-p65, p-IκBα, p-p38	[118]
*CRC*	HCT116 cells	60 µM, 48 h	apoptosis, caspase-3 and - 9		[119]
		100 µM, 24 h	cleavage of PARP, PPARγ	pro-caspase-3, -8, and -9, p-IκBα, iNOS, p50 (nuclear), p65 (nuclear), MMP-2 and -9	[120]
		40 µM, 4 h		p-Nrf2/Nrf2	[121]
		40 µmol/L, 24 or 48 h	p53, p21	Ezrin, cyclin D1, CDK4	[122]
	HCT116, A549, and Panc-1 cells	40 µM, 48 h	apoptotic rate, cleaved of caspase-3 and -9, DEPP, GADD45A, p-JNK, p-ERK, p-p38		[123]
	HCT116, SW480, and HT29 cells	20 µM, 72 h	apoptotic cells, hTERT, p-ERK, p-p38		[89]
	HT29 cells	100 µM, 48 h	cleavage of PARP and caspase-3, Bax, p53	Bcl-2, p-AKT, p-caspase- 9, p-GSK-3β, survivin, cyclin D1, cyclin B1	[124]
	HT29 xenograft	10 mg/kg three times/week, 43 days	apoptotic cells, p53, p21		[124]
	LS174T cells	25 µM, 48 h	PXR, CDX2		[100]
	DLD-1 cells	30 µM, 24 h		MMP-2 and -9, p-AKT	[125]
	HT29 and DLD1 cells	30 µM, 24 h		MMP-2 and -9, p-ERK	[126]
	DLD1 xenograft	20 mg/kg/day, 21 days		MMP-2 and -9, p-ERK	[126]
	HCT116 xenograft	50 mg/kg, 3 weekly cycle		Ki67	[122]
*Miscellaneous*	RAW 264.7cells (LPS-treated)	50 μM, 2 h or 48 h		TLR4, MyD88, IRAK-1, COX-2, NO, iNOS, p-JNK, p-ERK1/2, p-p38, MD-2/TLR4	[118]
		80 μM, 2 h	p-ERK, p-p38, p-JNK, p- JAK1, p-JAK2	iNOS, TNF-α, IL-6, IL-1β, NO, PGE2, p-STAT1(Tyr701), p-STAT3(Tyr705), p-STAT3(Ser727)	[127]
		10 μM, 2 h		TNF-α, COX-2, iNOS, NO, IL-1β, PGE2, p-IκBα	[128]
	RAW 264.7 cells (polyinosinic–polycytidylic-acid-treated)	100 µM, 24 h		NO, calcium release, IL-1α, IL-6, G-CSF, GM-CSF, VEGF, MCP-1, IP-10, LIX, RANTES, STAT1, LIX, STAT3, CHOP, Fas	[129]
	THP-1 cells (LPS-treated)	50 μM, 2 h		iNOS, COX-2, IL-1α, IL-1β, NLRP3, ASC, caspase-1	[118]
	EL-4 cells	50 μM, 24 h	CYP1A1, AHR (nuclear)	AHR (cytosol)	[117]
	colonic lamina propria lymphocyte	40 mg/kg, 10 days	CD4^+^CD25^+^Foxp3^+^ T cells	CD4^+^IL-17^+^ T cells	[117]
	mesenteric lymph of mice	40 mg/kg, 10 days	CD4^+^CD25^+^Foxp3^+^ T cells	CD4^+^IL-17^+^ T cells	[117]

AHR, aryl hydrocarbon receptor; AKT, protein kinase B; ALT, alanine aminotransferase; ASC, apoptosis-associated speck-like protein containing a caspase recruitment domain; AST, aspartate aminotransferase; ATP, adenosine triphosphate; Bax, Bcl-2-associated X protein; Bcl-2, B-cell lymphoma-2; cAMP, cyclic adenosine monophosphate; CD, Crohn’s disease; CDK4, cyclin-dependent kinase4; CDX2, caudal-type homeobox 2; cGMP, cyclic guanosine monophosphate; CHOP, C/EBP-homologous protein; COX-2, cyclooxygenase-2; CRC, colorectal cancer; CYP1A1, Cytochrome P450, family 1, subfamily A, polypeptide 1; DEPP, decidual protein induced by progesterone; DSS, dextran sulfate sodium; ERK, extracellular signal-regulated kinase; GADD45A, growth arrest and DNA-damage-inducible 45 alpha; G-CSF, granulocyte colony-stimulating factor; GGT, γ-glutamyl transpeptidase; GM-CSF, granulocyte macrophage colony-stimulating factor; GSK-3β, glycogen synthase kinase-3 beta; hTERT, human telomerase reverse transcriptase; ICAM-1, intercellular adhesion molecule-1; IFN-γ, interferon gamma; IKKβ, inhibitor of nuclear factor kappa-B kinase subunit beta; IL-, interleukin; iNOS, inducible nitric oxide synthase; IRAK-1, inhibition of interleukin-1 receptor-associated kinase 1; IκBα, nuclear factor of kappa light polypeptide gene enhancer in B-cells inhibitor, alpha; p-, phosphorylation; JAK, Janus protein tyrosine kinase; JNK, c- Jun N-terminal kinase; LIX, lipopolysaccharide-induced CXC chemokine; LPS, lipopolysaccharide; MCP-1, monocyte chemotactic protein-1; MDA, malondialdehyde; MDR1, multidrug resistance protein 1; MD-2, myeloid differentiation protein-2; MMP, matrix metalloproteinase; MyD88, myeloid differentiation factor 88; NF-κB, nuclear factor kappa-light-chain-enhancer of activated B cells; NO, nitric oxide; NLRP3, nucleotide-binding oligomerization domain-like receptor pyrin domain containing 3; Nrf2, nuclear factor erythroid-2-related factor 2; RANTES, regulated upon activation, expressed by normal T cell and presumably secreted; p-, phosphorylation; PARP, poly(ADP-ribose) polymerase; PGE2, prostaglandin E2; PPARγ, peroxisome proliferator-activated receptors γ; PXR, pregnane X receptor; RANTES, chemokine ligand 5; SOD, superoxide dismutase; STAT, signal transducer and activator of transcription; TNF, tumor necrosis factor; TLR4, toll-like receptor 4; TNBS, trinitrobenzene sulphonic acid; TG, triglyceride; VEGF, vascular endothelial growth factor; UC, ulcerative colitis.

**Table 3 ijms-24-01954-t003:** The effect of wogonoside on inflammatory bowel disease and colorectal cancer.

Disease	Animal/Cell Models	Dose and Times	Up-Regulation	Down-Regulation	Refs.
*UC*	DSS-induced UC mice	50 mg/kg, 10 days	occludin, ZO-1, Claudin1, MLC2	IL-13, IFN-γ, MLCK, p-MLC2	[130]
		50 mg/kg, 10 days	NF-κB p65 (cytosol)	MPO activity, iNOS activity, CD11b^+^F4/80^+^ monocyte/macrophage, CD11b^+^Gr-1^+^ neutrophils, IL-1β, TNF-α, and IL-6 (serum, colonic), IL-18, IFN-γ, and MIP-1α (colonic), NF-κB p65 (nucleus), p-IkBα (cytosol), p-p65, NF-kB DNA binding activity, cleaved caspase-1 and IL-1β, NLRP3, ASC, caspase-1 activity	[131]
	THP-1 cells (LPS-treated)	50 µM, 6 h	NF-κB p65 (cytosol)	IL-1β, IL-6, TNF-α, NF-κB p65 (nucleus), p-IkBα (cytosol), p-p65, NF-kB DNA binding activity, NLRP3, pro-caspase-1, cleaved caspase-1 and IL-1β, NLRP3, ASC, caspase-1 activity	[131]
	Caco-2 cells (TNF-α exposure)	50 µM, 72 h	occludin, ZO-1, Claudin1, TER, MLC2, F-actin	FD4, MLCK, p-MLC2	[130]
*CRC*	AOM/DSS mouse	100 mg/kg, 105 days	apoptotic cells, NF-κB (cytoplasm)	neutrophil (Gr-1^+^ positive cells), macrophage (F4/80^+^ positive cells), Ki-67, BrdU, PCNA, IL-1β, IL-6, TNF-α, NF-κB, NF-κB (nucleus), p-p65, PI3K, p-AKT/AKT, p-IKKα/IKKα, p-IkBα/ IkBα, cyclin D1, survivin	[132]
	HCT116 and HT29 cells (conditioned media from LPS-treated THP-1 cells)	150 µM, 24 h	NF-κB (cytoplasm)	PCNA, p-IKKα/IKKα, p-IκBα/IκBα, cyclin D1, survivin, NF-κB (nucleus), NF-κB fluorescence, NF-κB luciferase activity, PI3K, p-AKT/AKT	[132]
	LOVO cells	62.5 µM, 48 h	caspase-3 and -9, bax, LC3, p62	Bcl-2, PI3K, p-AKT, p-mTOR, p-p70S6K	[90]
*Miscellaneous*	RAW 264.7 cells (LPS-treated)	50 μM, 24 h		NO, PGE2, TNF-α, IL-6, iNOS, COX-2	[133]

AKT, protein kinase B; AOM, azoxymethane; ASC, apoptosis-associated speck-like protein containing a CARD (caspase recruitment domain); Bax, Bcl-2-associated X protein; BrdU, bromodeoxyuridine; COX-2, cyclooxygenase-2; CRC, colorectal cancer; DSS, dextran sulfate sodium; ERK, extracellular signal-regulated kinase; FD4, FITC-dextran; IFN-γ, interferon gamma; IL-, interleukin; IKKα, IκB kinase alpha; iNOS, inducible nitric oxide synthase; IκBα, nuclear factor of kappa light polypeptide gene enhancer in B-cells inhibitor alpha; JNK, c-Jun N-terminal kinase; LC3, microtubule-associated protein 1A/1B-light chain 3; LPS, lipopolysaccharide; MIP-1α, macrophage inflammatory proteins-1α; MLC2, myosin light chain2; MLCK, myosin light chain kinase; MPO, myeloperoxidase; mTOR, mammalian target of rapamycin; NF-κB, nuclear factor kappa-light-chain-enhancer of activated B cells; NO, nitric oxide; NLRP3, nucleotide-binding oligomerization domain-like receptor pyrin domain containing 3; p-, phosphorylation-; PCNA, proliferating cell nuclear antigen; PGE2, prostaglandin E2; PI3K, phosphoinositide 3-kinases; p70S6K, ribosomal protein S6 kinase; TER, transmembrane resistance; TNF-α, tumor necrosis factor-α; UC, ulcerative colitis; ZO-1, zonula occludens-1.

**Table 4 ijms-24-01954-t004:** The effect of wogonin on inflammatory bowel disease and colorectal cancer.

Disease	Animal/Cell Models	Dose and Times	Up-Regulation	Down-Regulation	Refs.
*UC*	DSS-induced UC mice	30 mg/kg, 7 days	GST, GSH, SOD, IL-10 (tissue), Bax, caspase-3 and -9, Nrf2, HO-1	MPO, NO, TBARS, TNF-α (serum, tissue), IL-6 (serum, tissue), PGE2 (tissue), Bcl-2, COX-2, iNOS, TLR4, p65	[134]
*CRC*	AOM/DSS mouse	100 mg/kg, 25 weeks	cleavage of caspase-3 and -9, IRE1α, PDI		[135]
		60 mg/kg, 105 days	IOD of Nrf2-positive cells (surrounding tissues), NF-kB p65 (cytoplasm), Nrf2 (nuclear)	IOD of BrdU-, PCNA-, and NF-kB-positive cells (surrounding tissues, tumor tissues), IOD of IL-6- and 1β-positive cells, IOD of Nrf2-positive cells (tumor tissues), NF-kB p65 (nuclear)	[136]
	HCT116 cells (cocultured with LPS-treated THP-1 cells)	100 µM, 12 h		IL-6, IL1β, PCNA, NF-kB (nuclear), p-IκB, p-IKK(α/β)	[136]
	HCT116 cells (LPS-treated)	50 µM, 8 h	Nrf2 (nuclear), NQO-1, HO-1	NF-kB (nuclear), p-IκB, p-IKKα/β, p-p38, p-ERK, p-JNK, p-AKT, PI3K, Nrf2 (cytoplasm)	[136]
	HCT116 cells	40 µM, 24 h	p21, p27, E2F1, p-Rb, GSK-3β, AXIN, p-β-catenin (Ser33/37/Thr41), apoptotic rate	cyclin A, cyclin E, cyclin D1, CDK2, CDK4, CDK8, Rb, Wnt3a, Dvl2, LRP6, p-GSK-3β, β-catenin, β-catenin (cytosolic), β-catenin (nuclear), TCF1, TCF3, TCF4, LEF1, c-Myc, survivin	[137]
		10 µM, 72 h	cleavage of caspase-3 and -9, apoptotic cells, IRE1α, calnexin, PDI, CHOP, DDIT3, XBP1, p53(Ser376), p53(Ser376) (cytoplasm), p53(Ser315)	colony formation rate, LC3-II, ATG12, p53(Ser376) (nuclear), p-p53(Ser15) (nuclear/ cytoplasm)	[135]
		40 µM, 24 h	p53, TIGAR, p-p53(Ser15), p-p53(Ser20), Ac-p53(Lys382)	glucose uptake, lactate generation, ATP production, PGM, GLUT1, HK2, PDHK1, LDHA, MDM2,	[138]
	HCT116 cells xenograft	60 mg/kg	E2F1	β-catenin, cyclin D1, c–Myc, TCF3	[137]
	HT29 cells	100 µM, 48 h	cleavage of PARP and caspase-3, Bax, p53,	Bcl-2, p-AKT, p-caspase-9, p-GSK-3β, survivin, cyclin D1, cyclin B1	[124]
	HT29 xenograft	10 mg/kg three times/week, 43 days	apoptotic cells, p53, p21		[124]
	SW48 cells	16 μM	LC3-II, cleavage of caspase-3, -8, -9, and PARP, PI3K	p-PI3K, p-AKT, p-STAT3(Tyr705), p-STAT3(Ser727)	[139]
	SW-480 cells	60 μM, 48 h	apoptotic cells, caspases-3, -8, and -9 activity		[96]
	THP-1 cells (LPS-treated)	100 µM	Nrf2 (nuclear), Keap1, HO-1	Nrf2 (cytosolic)	[136]
*Miscellaneous*	RAW 264.7 cells (LPS-treated)	100 µM, 24 h		NO, PGE2, iNOS, COX-2	[140]
		50 µM, 24 h		PGE2, COX-2, COX-2 activity	[141]
	DSS + MSCs	10 mg/kg, 8 days	IL-10	TNF-α	[142]
	MSCs + LPS	50 μM, 24 h	ROS, HIF-1α, IL-10, p-GSK-3β, p-GSK-3β/GSK-3β, p-AKT, p-AKT/AKT		[142]
	Caco-2 cells (LPS-treated)	50 μM	TEER, ZO-1, claudin-1, IκB	FD4, FD10, FD20, IL-1β, IL-6, IL-8, iNOS, COX-2, NF-κB p65, p-IκB, TLR4, MyD88, p-TAK1	[143]

Ac-, acetylation; AKT, protein kinase B; AOM, azoxymethane; ATG12, autophagy-related gene 12; ATP, adenosine triphosphate; Bax, Bcl-2-associated X protein; Bcl-2, B-cell lymphoma-2; CDK4, cyclin-dependent kinase4; COX-2, cyclooxygenase-2; CHOP, C/EBP-homologous protein; CRC, colorectal cancer; DDIT3(CHOP), DNA damage-inducible transcript 3; DSS, dextran sulfate sodium; Dvl2, disheveled; ERK, extracellular signal-regulated kinase; E2F1, E2F transcription factor 1; FD, fluorescein isothiocyanate-dextran; GLUT1, glucose transporter 1; GSK-3β, glycogen synthase kinase-3 beta; GSH, glutathione; GST, glutathione-s-transferase; HIF-1α, hypoxia-inducible factor 1-alpha; HK2, hexokinase 2; HO-1, heme oxygenase-1; IL-, interleukin; iNOS, inducible nitric oxide synthase; IOD, integrated optical density; IRE1α, inositol-requiring transmembrane kinase endoribonuclease-1 alpha; JNK, c-Jun N-terminal kinase; Keap1, Kelch-like ECH-associated protein 1; LC3-II, microtubule-associated protein 1A/1B-light chain 3-II; LDHA, lactate dehydrogenase A; LEF1, lymphoid enhancer factor1; LPS, lipopolysaccharide; LRP6, low-density lipoprotein receptor-related protein 6; MDM2, mouse double minute 2 homolog; MPO, myeloperoxidase; MSCs, mesenchymal stem cells; MyD88, myeloid differentiation primary response 88; NF-κB, nuclear factor kappa-light-chain-enhancer of activated B cells; NQO-1, NADPH quinone oxidoreductase-1; NO, nitric oxide; Nrf2, nuclear factor erythroid 2-related factor 2; p-, phosphorylation; PARP, poly(ADP-ribose) polymerase; PCNA, proliferating cell nuclear antigen; PDHK1, pyruvate dehydrogenase kinase 1; PDI, protein disulfide isomerase; PGE2, prostaglandin E2; PGM, phosphoglycerate mutase; PI3K, phosphoinositide 3-kinase; Rb, retinoblastoma; ROS, reactive oxygen species; SOD, superoxide dismutase; STAT3, signal transducer and activator of transcription3; TAK1, transforming growth factor beta-activated kinase 1; TBARS, thiobarbituric acid reactive substances; TCF, T cell factor; TEER, transepithelial electrical resistance; TIGAR, TP53-inducible glycolysis and apoptosis regulator; TLR4, toll-like receptor 4; TNF-α, tumor necrosis factor α; UC, ulcerative colitis; XBP1, X-box binding protein 1; ZO-1, zonula occludens-1.

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
