# Peer review of "Therapeutic Potential of Bioactive Components from Scutellaria baicalensis Georgi in Inflammatory Bowel Disease and Colorectal Cancer: A Review"

_ijms, 2023, doi:10.3390/ijms24031954_

Round 1
Reviewer 1 Report
Dear Editor,
thanks so much for the opportunity to revise the work entitled "Therapeutic Potential of Bioactive Components from Scutellaria baicalensis Georgi in Inflammatory Bowel Disease: A Review”.
The work is very interesting, showing the efficacy of SBG in IBD and CRC and may serve as a reference for future research and development of drugs for IBD and cancer treatment.
The paper is well written, the results are clearly reported and the review of the literature rigorous.
I have not specific revisions for the authors to perform. I only suggest improving the introduction with more information about Scutellaria baicalensis Georgi and other plants nutraceuticals, which are recommended in IBD.
Thanks.
Author Response
Response to Reviewer 1 Comments Dear editor Mr. Stephen Cong and reviewers: Thank you for your letter and for the reviewers’ comments concerning our manuscript entitled “Therapeutic Potential of Bioactive Components from Scutellaria baicalensis Georgi in Inflammatory Bowel Disease: A Review” (ID: ijms-2155274). These comments are all valuable and very helpful for revising and improving our paper, as well as the important guiding significance to our researches. We have studied comments carefully and have made corrections which we hope meet with approval. Revised portions are marked in red on the paper. The main corrections in the paper and the response to the reviewer’s comments are as flowing: • Point 1: I have not specific revisions for the authors to perform. I only suggest improving the introduction with more information about Scutellaria baicalensis Georgi and other plants nutraceuticals, which are recommended in IBD. Response 1: Thanks for your comment. We wrote about currently used treatments for IBD, but we did not think about other plants nutraceuticals that are effective for IBD. There are reports that many herbs, such as Ba Ji Tian, Qing Dai, and Dan Shen have some effectiveness in treating ulcerative colitis [1]. In addition, there are too many to include in our manuscript, so we won't add them. However, thank you for giving us the opportunity to investigate this one more time. [References for the responses] 1. Liu, Y.; Li, B.G.; Su, Y.H.; Zhao, R.X.; Song, P.; Li, H.; Cui, X.H.; Gao, H.M.; Zhai, R.X.; Fu, X.J.; et al. Potential activity of traditional chinese medicine against ulcerative colitis: A review. J. Ethnopharmacol. 2022, 289, 115084, doi:10.1016/j.jep.2022.115084. We tried our best to improve the manuscript and made some changes in the manuscript. These changes will not influence the content and framework of the paper. And here we did not list the changes but marked them in red in the revised paper. We appreciate for Editors/Reviewers’ warm work earnestly and hope that the correction will meet with approval. Once again, thank you very much for your comments and suggestions.
Reviewer 2 Report
Since the prevalence of IBD is rising it is extremely important to explore novel treatments for it. I have a few comments for the authors that may help make this review more appealing to readers.
line 48, 30 years of age maybe? please complete the sentence for it to be clear.
Something can be said about novel pharmacologic approaches to treatment of IBD in terms of adverse drug reactions, especially for JAK inhibitors as well and something can be said about IBS.
Can you please try to connect the proposed mechanisms of SBG with pathogenesis of IBD?
Also, InVivo and ExVivo models section can be omitted maybe or provide rationale why this is necessary.
A lot is said about CRC, you should consider mentioning it in the Title as well.
Author Response
Response to Reviewer 2 Comments Dear editor Mr. Stephen Cong and reviewers: Thank you for your letter and for the reviewers’ comments concerning our manuscript entitled “Therapeutic Potential of Bioactive Components from Scutellaria baicalensis Georgi in Inflammatory Bowel Disease: A Review” (ID: ijms-2155274). These comments are all valuable and very helpful for revising and improving our paper, as well as the important guiding significance to our researches. We have studied comments carefully and have made corrections which we hope meet with approval. Revised portions are marked in red on the paper. The main corrections in the paper and the response to the reviewer’s comments are as flowing: • Point 1: line 48, 30 years of age maybe? please complete the sentence for it to be clear. Response 1: Thanks for your comment. We don't mean about 30 years old in line 48. This indicates that the risk of CRC in IBD patients may increase by 18% after 30 years. • Point 2: Something can be said about novel pharmacologic approaches to treatment of IBD in terms of adverse drug reactions, especially for JAK inhibitors as well and something can be said about IBS. Response 2: Thanks for your suggestion. We know that Tofacitinib, an oral non-selective JAK inhibitor, was approved by the US FDA for the treatment of UC in May 2018 [1]. However, one of the major concerns with the use of JAK inhibitors is the reported risk of thromboembolic events [2]. We mentioned JAK inhibitors on line 125 of the manuscript. However, if we write down all the positive effects and side effects of each immunomodulator related to IBD as well as IBS treatment in the manuscript, it would be too much to explain, so we omitted it. We therefore ask for your understanding. • Point 3: Can you please try to connect the proposed mechanisms of SBG with pathogenesis of IBD? Response 3: Thank you for your comments. We noted that the exact cause of IBD is unclear (Line 80). Therefore, it is unlikely that we can directly link the proposed mechanism of action of SBG we mentioned to the pathogenesis of IBD. However, it is thought that the anti-inflammatory action of SBG suppresses the inflammatory response associated with the pathogenesis of IBD. • Point 4: Also, InVivo and ExVivo models section can be omitted maybe or provide rationale why this is necessary. Response 4: Thanks for your advice. We discussed the effects of the main active substances of SBG on IBD and CRC according to several models. Therefore, we believe that we can help readers understand by explaining the InVivo model of IBD. Also, we think that the current research direction of the ExVivo model can also provide good information to the readers. • Point 5: A lot is said about CRC, you should consider mentioning it in the Title as well. Response 5: Thanks for the advice. We have explained a lot about CRC as well as IBD. Therefore, following your advice, we have revised the manuscript by adding CRC to the title. [References for the responses] 1. Grossberg, L.B.; Papamichael, K.; Cheifetz, A.S. Review article: emerging drug therapies in inflammatory bowel disease. Aliment. Pharmacol. Ther. 2022, 55, 789-804, doi:10.1111/apt.16785. 2. Dudek, P.; Fabisiak, A.; Zatorski, H.; Malecka-Wojciesko, E.; Talar-Wojnarowska, R. Efficacy, safety and future perspectives of JAK inhibitors in the IBD treatment. J. Clin. Med. 2021, 10, doi:10.3390/jcm10235660. We tried our best to improve the manuscript and made some changes in the manuscript. These changes will not influence the content and framework of the paper. And here we did not list the changes but marked them in red in the revised paper. We appreciate for Editors/Reviewers’ warm work earnestly and hope that the correction will meet with approval. Once again, thank you very much for your comments and suggestions.